# Addressing Missing and Noisy Modalities in One Solution: Unified Modality-Quality Framework for Low-quality Multimodal Data

## Abstract

Multimodal data encountered in real-world scenarios are typically of low quality, with noisy modalities and missing modalities being typical forms that severely hinder model performance and robustness. However, prior works often handle noisy and missing modalities separately. In contrast, we jointly address missing and noisy modalities to enhance model robustness in low-quality data scenarios. We regard both noisy and missing modalities as a unified low-quality modality problem, and propose a unified modality-quality (UMQ) framework to enhance low-quality representations for multimodal affective computing. Firstly, we train a quality estimator with explicit supervised signals via a rank-guided training strategy that compares the relative quality of different representations by adding a ranking constraint, avoiding training noise caused by inaccurate absolute quality labels. Then, a quality enhancer for each modality is constructed, which uses the sample-specific information provided by other modalities and the modality-specific information provided by the defined modality baseline representation to enhance the quality of unimodal representations. Finally, we propose a quality-aware mixture-of-experts module with particular routing mechanism to enable multiple modality-quality problems to be addressed more specifically. UMQ consistently outperforms state-of-the-art baselines on multiple datasets under the settings of complete, missing, and noisy modalities.

## 1 Introduction

Real-world entities and events are inherently described through heterogeneous modalities, and humans perceive the environment via multiple sensory channels (e.g., visual and auditory). This observation highlights the critical necessity of fusing information from disparate sources (Baltrušaitis et al., 2019). Multimodal affective computing (MAC) (Poria et al., 2017; Guo et al., 2025), which systematically integrates linguistic, acoustic, and visual signals emitted by speakers to jointly model and predict human sentiment, opinion, mental state, and intent, thus constitutes a pivotal and rapidly evolving direction in multimodal learning with substantial translational potential.

Nevertheless, in real-world scenarios, multimodal data frequently exhibit degraded quality, manifesting primarily as modality-specific noise and incompleteness. Such impairments substantially degrade model performance, compromise robustness, and constrain practical applicability (Zhang et al., 2024). As shown in Figure 1, missing modality typically arises from unavailable acquisition equipment or

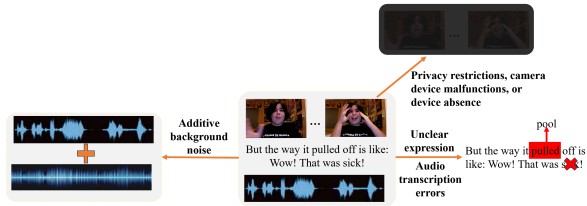

Figure 1: An example of missing and noisy modalities.

sensor failures, whereas noisy modality originates from background interference, inherent sensor inaccuracies, data transmission artifacts, etc. Current works address missing and noisy modalities separately (Guo et al., 2024; Wang et al., 2023). However, since both issues are common and often occur simultaneously, separate handling limits the application scope and robustness of the model.

Consequently, we propose a framework to **jointly handle missing and noisy modalities**, aiming to enhance model robustness in low-quality data scenarios. We regard missing modality as a special type of noisy modality, where the noise generation pattern and affected modality are known. Then, we propose a unified modality-quality (UMQ) framework to simultaneously handle multiple low-quality data problems. UMQ reframes both noisy and missing modalities as a unified low-quality modality problem, systematically enhancing modality representations through three synergistic components: (i) a **quality estimator** that quantifies representation fidelity, (ii) a **quality enhancer** that leverages modality-specific and sample-specific priors to recover informative features, and (iii) a **modality-quality-aware mixture-of-experts (MQ-MOE)** architecture that adaptively handles enhanced representations under varying quality conditions.

Specifically, compared to previous works that do not define explicit labels for the learning of unimodal weights or confidences (Li et al., 2024b), we train a quality estimator for each modality in an explicit supervised manner to more accurately identify the quality of the modality. As absolute labels for modality quality are hard to determine, we propose a **rank-guided training strategy** and define multiple types of relative labels for quality learning. The proposed training strategy compares the relative quality of different modality representations by adding a ranking constraint, avoiding training noise caused by inaccurate absolute labels. It enables more flexible and accurate learning of modality quality, as it does not enforce the estimated quality scores to fit an absolute value.

Afterwards, unlike prior approaches that directly reconstruct missing modalities using available modalities (Lian et al., 2023; Shi et al., 2025), we train a quality enhancer for each modality that uses the '**sample-specific information**' provided by other modalities and '**modality-specific information**' provided by the defined modality baseline representation to enhance the quality of unimodal representations, avoiding the problem that the generated representations do not contain modality-specific information. The established modality baseline representation captures global distribution and inherent properties of the corresponding modality, supplying modality-specific information for low-quality representations and enhancing their fidelity. Leveraging (1) the modality-specific baseline representation (i.e., modality-level characteristics independent of samples) and (2) sample-specific information from other modalities, quality enhancer augments representations while ensuring modality-specific details are learned. By integrating modality-level and sample-level cues, quality enhancer can generate richer, higher-quality modality representations.

Finally, we propose the MQ-MoE architecture that **enables a unified framework to handle diverse modality-missing and modality-noise configurations more specifically**. For $|\mathcal{M}|$ input modalities, each can be either high- or low-quality, yielding $2^{|\mathcal{M}|}$ distinct quality combinations. A single shared predictor becomes impractical for such combinatorial explosion, especially as $|\mathcal{M}|$ increases. To address this, MQ-MoE employs specialized expert modules that separately process each modality-quality combination, ensuring dedicated handling of every possible configuration. Moreover, we enforce several constraints on expert selection to ensure that samples sharing the same modality-quality configuration (i.e., identical language, acoustic, and visual quality levels) are routed to similar experts, whereas samples with different configurations activate distinct experts.

The main contributions of this paper are listed as below:

- We innovatively address noisy modality and missing modality issues in a unified framework, enhancing model robustness in real-world scenarios. UMQ consistently outperforms state-of-the-art methods on multiple MAC datasets under the settings of complete modalities, missing modalities, and noisy modalities.

- We devise the quality estimator that quantifies representation quality of each modality, and propose a rank-guided optimization strategy to compare the relative quality between different representations to more accurately train the quality estimator in a supervised way, avoiding training noise caused by inaccurate absolute labels.

- We propose the quality enhancer that use 'sample-specific information' provided by other modalities and 'modality-specific information' provided by the proposed modality baseline representation to enhance the quality of unimodal representations, ensuring that the generated representations can contain modality-specific information.

- We introduce MQ-MOE that enables a unified framework to handle diverse modality-missing and modality-noise configurations, and enforce several constraints to ensure that samples sharing the same modality-quality configuration are routed to similar experts.

## 2 RELATED WORK

### 2.1 MISSING MODALITY

The works that handle missing modality problem can be roughly categorized into four categories: (1) **Data augmentation methods** simulate the absence of modality during training to improve generalizability via modality drop, noise inference, etc (Lin & Hu, 2024; Hazarika et al., 2022). However, they usually lead to a decrease in the performance under complete modalities; (2) **Alignment-based methods** align the features of incomplete and complete modalities through contrastive learning (Poklukar et al., 2022), canonical correlation analysis (Andrew et al., 2013; Sun et al., 2020), or knowledge distillation (Li et al., 2024c; Zhong et al., 2025), etc. But they typically fail to maintain the performance of trained models due to the lack of modality recovery mechanisms; (3) **Reconstruction-based methods** recover missing modality features using generative models, such as autoencoders (Tran et al., 2017; Zeng et al., 2022), graph networks (Lian et al., 2023), diffusion models (Wang et al., 2023), and prompt-tuning strategies (Guo et al., 2024). They usually directly use existed modalities to reconstruct missing modalities, and the generated features often fail to convey modality-specific information. In contrast, we comprehensively leverage modality-level and sample-level cues to enable the enhanced representations to contain modality-specific information; (4) **Architecture-based methods** leverage architectures that can naturally handle any number of modalities (attention networks, ensemble frameworks, etc) to address missing modalities (Deng et al., 2025; Xue & Marculescu, 2023). For example, Xu et al. (2024a) apply MOE for unimodal expert training and experts mixing training, aiming to learn discriminative unimodal and joint representations. But they overlook handling various modality-missing/-noise scenarios. Differently, UMQ leverages the power of MOE to jointly handle multiple modality-missing/-noise situations, and designs quality-aware routing mechanisms to address each situation specifically.

### 2.2 NOISY MODALITY

The works that handle modality noise mainly fall into three categories: (1) **Noise augmentation methods** simulate real-world noise by adding noise to features or raw inputs and designing defense mechanisms accordingly (Hazarika et al., 2022; Mao et al., 2023). However, designing defense mechanisms for each type of noise is complex and has limited generalizability; (2) **Representation regularization methods** use techniques such as tensor rank minimization and information bottleneck to extract discriminative representations from noisy features (Liang et al., 2019; Mai et al., 2023b; Federici et al., 2020). However, they rely on assumptions that may not hold in real-world scenarios; (3) **Noise identification and filtering methods** aim to identify and suppress noisy information using attention/gating mechanisms (Gong et al., 2025; Xue et al., 2023; Gao et al., 2024). However, they do not define explicit supervised labels for the learning of attention weights/confidences. In contrast, we train the quality estimator with explicit labels, and propose the rank-guided training strategy to leverage relative quality labels for the learning of modality quality.

In addition, **the joint handling of noisy and missing modalities remains an open challenge**. As these two forms of data degradation frequently co-occur in the real-world, we jointly resolve them to broaden the practical applicability of the model.

## 3 ALGORITHM

The diagram of UMQ and the training objectives of the proposed quality enhancer, modality decoupling operation, quality estimator, and the MQ-MoE are shown in Figure 2. UMQ is evaluated on multiple MAC tasks, including multimodal sentiment analysis (MSA) (Zadeh et al., 2016), multimodal humor detection (MHD) (Hasan et al., 2019), and multimodal sarcasm detection (MSD) (Castro et al., 2019). The input is a video segment described by acoustic ($a$), visual ($v$), and language ($l$) modalities. The input feature sequences are denoted as $\{\boldsymbol{U}_m \in \mathbb{R}^{T_m \times d_m} | m \in \{a, v, l\}\}$, where $T_m$ is the sequence length and $d_m$ is the feature dimensionality. We apply unimodal networks to extract unimodal representations $\boldsymbol{x}_m \in \mathbb{R}^{1 \times d}$ based on $\boldsymbol{U}_m$ (the structures of unimodal networks are introduced in Section B.1). If a modality is absent, we use Gaussian noise as its default features.

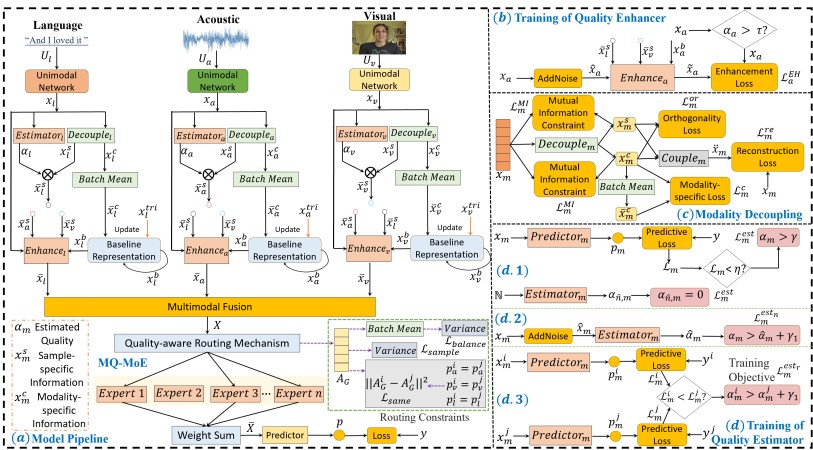

Figure 2: Diagram of UMQ and the training objectives of the proposed components.

## 3.1 QUALITY ESTIMATION AND ENHANCEMENT

Given unimodal representation $\boldsymbol{x}_m \in \mathbb{R}^{1 \times d}$, we establish a $Sigmoid$-activated **quality estimator** $Estimator_m$ to generate the estimated quality score $\alpha_m \in \mathbb{R}^1$ for $\boldsymbol{x}_m$:

$$\alpha_m = Estimator_m(\boldsymbol{x}_m; \theta_{E_m}) \tag{1}$$

Then, we enhance the quality of unimodal representations using 'sample-specific information' and 'modality-specific information'. We first illustrate how we obtain sample-specific and modality-specific information via the proposed **modality decoupling operation**:

$$\boldsymbol{x}_m^c, \ \boldsymbol{x}_m^s = Decouple_m(\boldsymbol{x}_m; \ \theta_{decouple_m}); \quad \mathcal{L}_m^{or} = ||Nor(\boldsymbol{x}_m^s)(Nor(\boldsymbol{x}_m^c))^R||_2 \tag{2}$$

$$\mathcal{L}_m^{MI} = max(||\boldsymbol{x}_m - \boldsymbol{x}_m^c||^2 - \epsilon, 0) + max(||\boldsymbol{x}_m - \boldsymbol{x}_m^s||^2 - \epsilon, 0) \tag{3}$$

$$\mathcal{L}_m^c = \frac{1}{n} \sum_{i=1}^n ||(\boldsymbol{x}_m^c)_i - \frac{1}{n} \sum_{j=1}^n (\boldsymbol{x}_m^c)_j||^2 \tag{4}$$

$$\ddot{\boldsymbol{x}}_m = Couple_m(\boldsymbol{x}_m^s, \boldsymbol{x}_m^c; \ \theta_{couple_m}); \quad \mathcal{L}_m^{re} = ||\boldsymbol{x}_m - \ddot{\boldsymbol{x}}_m||_2 \tag{5}$$

where $Nor$ denotes the L2 normalization operation, $Decouple_m$ and $Couple_m$ denote the decouple and couple networks for modality $m$, $\boldsymbol{x}_m^s$ and $\boldsymbol{x}_m^c$ are the sample-specific and sample-shared (modality-specific) representations for modality $m$, $n$ is the batch size ($i$ and $j$ are the indices of samples), $\epsilon$ is the maximum distance that ensures $\boldsymbol{x}_m^c/\boldsymbol{x}_m^s$ and the original representation $\boldsymbol{x}_m$ maintain a degree of similarity but are not identical, and $\ddot{\boldsymbol{x}}_m$ is the reconstructed representation. The orthogonality loss $\mathcal{L}_m^{or}$ in Eq. 2 enforces orthogonality between $\boldsymbol{x}_m^s$ and $\boldsymbol{x}_m^c$. The mutual information constraint $\mathcal{L}_m^{MI}$ in Eq. 3 ensures that $\boldsymbol{x}_m^c/\boldsymbol{x}_m^s$ maintains relevance to $\boldsymbol{x}_m$, thus preventing $\boldsymbol{x}_m^c/\boldsymbol{x}_m^s$ from only containing information unrelated to $\boldsymbol{x}_m$. The modality-specific loss $\mathcal{L}_m^c$ in Eq. 4 encourages that $\boldsymbol{x}_m^c$ remains the same for all samples. The reconstruction loss $\mathcal{L}_m^{re}$ in Eq. 5 forces $\boldsymbol{x}_m^s$ and $\boldsymbol{x}_m^c$ to fully encompass the information in $\boldsymbol{x}_m$, preventing information loss during decomposition. The total loss for the modality decoupling operation $\mathcal{L}^{de}$ is then defined as:

$$\mathcal{L}^{de} = \sum_m \mathcal{L}_m^c + \mathcal{L}_m^{MI} + \mathcal{L}_m^{re} + \mathcal{L}_m^{or} \tag{6}$$

After obtaining sample-specific representation $\boldsymbol{x}_m^s$ and modality-specific representation $\boldsymbol{x}_m^c$, we use $\boldsymbol{x}_m^c$ to construct **modality baseline representation** $\boldsymbol{x}_m^b$ that reveals the general distributional information of modality $m$, aiming to provide modality-specific information for quality enhancement:

$$\boldsymbol{x}_m^b \longleftarrow \frac{1}{2}(\lambda \times \boldsymbol{x}_m^b + (1 - \lambda) \times \frac{1}{n} \sum_{j=1}^n (\boldsymbol{x}_m^c)_j) + \frac{1}{2}\boldsymbol{x}_m^{tri} \tag{7}$$

where $\boldsymbol{x}_m^{tri} \in \mathbb{R}^{1 \times d}$ is a trainable embedding for modality $m$, $\lambda$ is a hyperparameter for moving average that is between 0 and 1. Notably, in Eq. 7, except for the regular moving average (the left part), we use the trainable embedding $\boldsymbol{x}_m^{tri}$ to construct modality baseline representation. This is because in practice, we can only select a small proportion of samples for each updating due to the

memory limitation of hardware, which inevitably introduces noise. Moreover, the features of samples change across iterations, and it might be difficult to balance the weights of previous embedding and the current one. Therefore, we learn a bias embedding $\boldsymbol{x}_m^{tri}$ that automatically compensates for the possible noise and error introduced by regular moving average.

Combining the **modality-specific and sample-specific information**, the **quality enhancer** aims to generate higher-quality unimodal representation $\bar{\boldsymbol{x}}_m$:

$$\bar{\boldsymbol{x}}_m = EH_m(\boldsymbol{x}_m, \{\boldsymbol{x}_{m'}^s \cdot \alpha_{m'} | m' \neq m\}, \boldsymbol{x}_m^b; \theta_{EH_m}) \tag{8}$$

where $EH_m$ is the abbreviation of quality enhancer $Enhancer_m, m' \in \{a, v, l\}, m' \neq m$. The $\boldsymbol{x}_{m'}^s$ is multiplied by its estimated quality $\alpha_{m'}$ so that the quality enhancer can be aware of the quality of modality $m'$ when leveraging its modality-specific information to enhance $\boldsymbol{x}_m$. Compared to prior approaches that directly reconstruct missing modalities using available modalities (Lian et al., 2023; Shi et al., 2025), the proposed quality enhancer comprehensively leverages the 'sample-specific information' provided by other modalities and 'modality-specific information' provided by modality baseline representation to enhance the quality of unimodal representations, avoiding the problem that the generated representations do not contain modality-specific information.

Then, we generate multimodal representation $\boldsymbol{X} \in \mathbb{R}^{1 \times d}$ based on the enhanced representations:

$$\boldsymbol{X} = Fusion(\bar{\boldsymbol{x}}_l, \bar{\boldsymbol{x}}_a, \bar{\boldsymbol{x}}_v; \theta_f) \tag{9}$$

In practice, we use a multi-layer perception (MLP) network as the fusion network $Fusion$.

### 3.2 MODALITY-QUALITY-AWARE MIXTURE-OF-EXPERTS

Each modality can be either high- or low-quality, yielding $2^{|\mathcal{M}|}$ distinct quality combinations ($|\mathcal{M}|$ denotes the number of modalities). A single shared predictor becomes impractical for such combinatorial explosion. Therefore, taking inspiration from the MOE framework (Riquelme et al., 2021; Mustafa et al., 2022), we design a MQ-MoE architecture that **enables a unified framework to specifically handle diverse modality-missing and modality-noise configurations**. MQ-MoE employs specialized expert modules and **quality-aware routing mechanism** that separately processes each modality-quality combination, ensuring dedicated handling of every possible configuration:

$$\boldsymbol{A}_G = \frac{\boldsymbol{W}_G(\boldsymbol{X})^R}{\sqrt{d}} \in \mathbb{R}^{h \times 1} \tag{10}$$

$$\hat{\boldsymbol{A}}_G, \mathbb{I} = Top_k(\boldsymbol{A}_G); \quad \bar{\boldsymbol{X}} = \sum_{j=1}^{k} Softmax(\hat{\boldsymbol{A}}_G)_j E_j(\boldsymbol{X}) \tag{11}$$

where $\boldsymbol{W}_G \in \mathbb{R}^{h \times d}$ is the gate control parameter ($h$ is the total number of experts), $\boldsymbol{A}_G$ is the expert selection vector (the experts corresponding to the top-$k$ largest values in $\boldsymbol{A}_G$ will be selected as the processing experts for $\boldsymbol{X}$), $\hat{\boldsymbol{A}}_G$ denotes the vector consisting of the top-$k$ largest values in $\boldsymbol{A}_G$ ($\mathbb{I}$ denotes their positions in $\boldsymbol{A}_G$), $E_j$ denotes the $j^{th}$ selected expert ($\{E_1, E_2, ..., E_k\} = \{E_j | j \in \mathbb{I}\}$).

MQ-MOE enforces several constraints on expert selection to ensure that samples sharing the same modality-quality configuration (i.e., identical language, acoustic, and visual quality levels) are routed to similar experts, whereas samples with differing configurations activate distinct experts:

$$\bar{\boldsymbol{A}}_G = \frac{1}{n} \sum_{i=1}^{n} \boldsymbol{A}_G^i; \quad \mathcal{L}_{balance} = Var(\bar{\boldsymbol{A}}_G) \tag{12}$$

$$\mathcal{L}_{sample} = \frac{1}{n} \sum_{i=1}^{n} max(0, -Var(\boldsymbol{A}_G^i) + \beta) \tag{13}$$

$$\mathcal{L}_{same} = \sum_{i,j} ||\boldsymbol{A}_G^i - \boldsymbol{A}_G^j||^2, \ s.t. \ p_m^i = p_m^j, \ \forall m \in \{l, a, v\} \tag{14}$$

where $n$ denotes the batch size, $Var$ is the variance function, $p_m^i$ denotes the quality level for modality $m$ in sample $i$ ($p_m^i = 0$ iff $\alpha_m^i < \tau$ else $p_m^i = 1$, where $\tau$ is a quality threshold hyperparameter). $\mathcal{L}_{balance}$ enforces equal selection probabilities across all experts at the batch level, preventing any subset of experts from being disproportionately activated. $\mathcal{L}_{sample}$ imposes that the variance of the expert-selection vector within each individual sample be at least $\beta$, thereby preventing uniform probabilities and enabling the router to accurately identify experts relevant to the given sample, $\mathcal{L}_{same}$

constrains samples that share an identical modality-quality configuration (i.e., both samples present the same modality-wise low- or high-quality patterns) to select the same set of experts. Collectively, these losses render the proposed MQ-MOE quality-aware: distinct experts address distinct modality-quality issues, while identical experts handle similar issues. Consequently, a unified framework can address a broad spectrum of modality-quality problems more effectively with greater specificity.

Finally, we construct a predictor to infer the final prediction $p$ and compute the predictive loss $\mathcal{L}$:

$$p = Predictor(\bar{\boldsymbol{X}}; \theta_{pre}); \quad \mathcal{L} = Loss(p, y) \tag{15}$$

where $y$ is the true label (the concrete form of $Loss$ depends on downstream tasks).

## 3.3 TRAINING OF QUALITY ESTIMATOR

Compared to prior works that do not define explicit labels for the learning of unimodal weights/confidences (Li et al., 2024b), we learn the quality estimator in an explicit supervised manner to more accurately identify modality quality. To generate precise supervised signals, we first discriminate between highest- and lowest-quality unimodal representations and assign them corresponding quality labels. Lowest-quality instances are simulated by Gaussian noise and receive a quality label of 0. Highest-quality instances are defined as unimodal representations whose predictive loss falls below a predefined threshold $\eta$, and are assigned a large label (larger than 0.95). We then leverage their accurate supervised signals to train the quality estimator:

$$\alpha_{\hat{n},m} = Estimator_m(\mathbb{N}; \theta_{E_m}); \quad \mathcal{L}_m^{est} = ||\alpha_{\hat{n},m}||^2 \tag{16}$$

$$\mathcal{L}_m^{est} \longleftarrow \mathcal{L}_m^{est} + max(0, \gamma - \alpha_m), \quad s.t. \, \mathcal{L}_m = Loss(Predictor_m(\boldsymbol{x}_m; \theta_m), y) < \eta \tag{17}$$

where $\mathbb{N} \in \mathbb{R}^{1 \times d}$ denotes the Gaussian noise, $\mathcal{L}_m$ denotes the unimodal predictive loss for modality $m$, $\eta$ is a threshold hyperparameter that is set to 0.01, and $\gamma$ is a hyperparameter that is set to 0.95.

However, for representations whose quality lies between these two extremes, assigning an absolute label is non-trivial, and defining conventional absolute labels (such as transforming predictive losses as the absolute labels (Mai et al., 2024)) would introduce label noise. Hence, we adopt a **rank-guided training strategy**: the estimator is trained to compare the relative quality of representations via a ranking loss that is constructed according to their relative unimodal predictive losses, enabling accurate ordinal quality judgments without relying on absolute labels:

$$\mathcal{L}_m^{est_r} = max(0, \, \alpha_m^i + \gamma_1 - \alpha_m^j), \, s.t. \, \mathcal{L}_m^j < \mathcal{L}_m^i \tag{18}$$

where $\gamma_1$ is the margin. Additionally, we construct an 'AddNoise' function to simulate real-world noise by randomly adding Gaussian noise to the original unimodal representations (features are replaced with Gaussian noise in the most severe cases), and apply the rank-guided training strategy to encourage the estimated quality of original representations to be larger than the corrupted ones:

$$\hat{\boldsymbol{x}}_m = AddNoise(\boldsymbol{x}_m); \quad \hat{\alpha}_m = Estimator_m(\hat{\boldsymbol{x}}_m; \theta_{E_m}) \tag{19}$$

$$\mathcal{L}_m^{est_n} = max(0, \hat{\alpha}_m + \gamma_1 - \alpha_m) \tag{20}$$

The total training loss for quality estimator is defined as:

$$\mathcal{L}^{est} = \sum_m \mathcal{L}_m^{est} + \mathcal{L}_m^{est_n} + \mathcal{L}_m^{est_r} + \mathcal{L}_m \tag{21}$$

## 3.4 TRAINING OF QUALITY ENHANCER

To enable quality enhancer to produce higher-quality features, we apply 'AddNoise' function to generate corrupted unimodal representations $\hat{\boldsymbol{x}}_m$, and use quality enhancer to enhance the quality of corrupted representations. In particular, to ensure more accurate training of the quality enhancer, we select high-quality unimodal representations (whose estimated quality score $\alpha_m$ is greater than the threshold $\tau$) to perform feature enhancement. The loss for quality enhancer $\mathcal{L}^{EH}$ is computed as:

$$\tilde{\boldsymbol{x}}_m = EH_m(\hat{\boldsymbol{x}}_m, \{\boldsymbol{x}_{m'}^s \cdot \alpha_{m'} | m' \neq m\}, \boldsymbol{x}_m^b; \theta_{EH_m}); \quad \mathcal{L}^{EH} = \sum_m ||\tilde{\boldsymbol{x}}_m - \boldsymbol{x}_m||^2, \, s.t. \, \alpha_m > \tau \tag{22}$$

## 3.5 MODEL OPTIMIZATION

We jointly optimize the predictive loss and the auxiliary losses, and the final loss $\mathcal{L}$ is defined as :

$$\mathcal{L} = \mathcal{L}_p + \beta_{de} \cdot \mathcal{L}^{de} + \beta_{est} \cdot \mathcal{L}^{est} + \beta_{EH} \cdot \mathcal{L}^{EH} + \beta_{moe} \cdot (\mathcal{L}_{balance} + \mathcal{L}_{sample} + \mathcal{L}_{same}) \tag{23}$$

Table 1: The results on CMU-MOSI and CMU-MOSEI under complete modalities. The results labeled with $^{\dagger}$ are obtained from original papers, and other results are obtained from our experiments.

| | CMU-MOSI | | | | | CMU-MOSEI | | | | |
|---|---|---|---|---|---|---|---|---|---|---|
| | Acc7↑ | Acc2↑ | F1↑ | MAE↓ | Corr↑ | Acc7↑ | Acc2↑ | F1↑ | MAE↓ | Corr↑ |
| MFM (Tsai et al., 2019) | 33.3 | 80.0 | 80.1 | 0.948 | 0.664 | 50.8 | 83.4 | 83.4 | 0.580 | 0.722 |
| DEVA$^{\dagger}$ (Wu et al., 2025) | 46.3 | 86.3 | 86.3 | 0.730 | 0.787 | 52.3 | 86.1 | 86.2 | 0.541 | 0.769 |
| AtCAF$^{\dagger}$ (Huang et al., 2025) | 46.5 | 88.6 | 88.5 | 0.650 | 0.831 | **55.9** | 87.0 | 86.8 | 0.508 | 0.785 |
| DLF$^{\dagger}$ (Wang et al., 2025) | 47.1 | 85.1 | 85.0 | 0.731 | 0.781 | 53.9 | 85.4 | 85.3 | 0.536 | 0.764 |
| C-MIB (Mai et al., 2023b) | 47.7 | 87.8 | 87.8 | 0.662 | 0.835 | 52.7 | 86.9 | 86.8 | 0.542 | 0.784 |
| ITHP (Xiao et al., 2024) | 47.7 | 88.5 | 88.5 | 0.663 | 0.856 | 52.2 | 87.1 | 87.1 | 0.550 | 0.792 |
| Multimodal Boosting (Mai et al., 2024) | 49.1 | 88.5 | 88.4 | 0.634 | 0.855 | 54.0 | 86.5 | 86.5 | 0.523 | 0.779 |
| UMQ | **49.7** | **90.1** | **90.0** | **0.630** | **0.863** | 55.5 | **88.1** | **88.1** | **0.506** | **0.796** |

where $\mathcal{L}_p$ is the predictive loss, $\beta_{de}$, $\beta_{est}$, $\beta_{EH}$ and $\alpha_{moe}$ are the weights of the auxiliary losses.

# 4 EXPERIMENTS

UMQ is evaluated on CMU-MOSI (Zadeh et al., 2016), CMU-MOSEI (Zadeh et al., 2018), CH-SIMS (Yu et al., 2020), UR-FUNNY (Hasan et al., 2019), and MUStARD (Castro et al., 2019) datasets. Due to space limitation, the experimental settings, baselines, datasets, and additional results (including hyperparameter analysis, results on CH-SIMS, etc) are introduced in the Appendix.

## 4.1 PERFORMANCE IN MSA UNDER COMPLETE MODALITIES

The results in the MSA task are presented in Table 1. On CMU-MOSI, UMQ outperforms strong baselines Multimodal Boosting and AtCAF in all metrics. On CMU-MOSEI, UMQ obtains the best results in Acc2, F1 score, MAE, and Corr. Notably, due to space limitation, we present the results on CH-SIMS (Yu et al., 2020) dataset in Section A.1, and the results also suggest that UMQ outperforms all baselines. Generally, **UMQ achieves state-of-the-art results in MSA even in the settings of complete modalities**. This is mainly because we learn the quality estimator and enhancer with explicit supervised signals, which can enhance unimodal representations. Moreover, MQ-MoE can effectively handle different modality-quality configurations, specifically addressing different types of inputs and thus boosting the performance of complex multimodal systems.

Table 2: The results on UR-FUNNY dataset.

| Model | Acc | Parameters |
|---|---|---|
| HKT$^{\dagger}$ (Hasan et al., 2021) | 77.4 | - |
| DMD+SuCI$^{\dagger}$ (Yang et al., 2024) | 70.8 | - |
| AtCAF$^{\dagger}$ (Huang et al., 2025) | 72.1 | - |
| HKT (Hasan et al., 2021) | 76.5 | 17,066,564 |
| MCL (Mai et al., 2023a) | 77.7 | 13,762,973 |
| MGCL (Mai et al., 2023c) | 78.1 | 14,062,342 |
| UMQ | **78.8** | **13,315,778** |

## 4.2 RESULTS IN MHD AND MSD UNDER COMPLETE INPUTS

To validate the generalizability of UMQ, we evaluate it in the MHD and MSD tasks using the UR-FUNNY and MUStARD datasets. Tables 2 and 3 show that UMQ surpasses the best existing methods, MGCL and MO-Sarcation, respectively, while maintaining a moderate parameter count. Thus, **UMQ reaches state-of-the-art performance with restrained complexity**, confirming its broad applicability to diverse multimodal learning tasks.

Table 3: The results on MUStARD dataset.

| Model | Acc | Parameters |
|---|---|---|
| HKT$^{\dagger}$ (Hasan et al., 2021) | 79.4 | - |
| MO-Sarcation$^{\dagger}$ (Tomar et al., 2023) | 79.7 | - |
| HKT (Hasan et al., 2021) | 76.5 | 17,101,372 |
| MCL (Mai et al., 2023a) | 77.9 | **13,828,449** |
| MGCL (Mai et al., 2023c) | 77.9 | 14,282,000 |
| UMQ | **80.6** | 14,362,506 |

## 4.3 DISCUSSION ON MISSING MODALITIES

For incomplete inputs, we compare UMQ with CIDer (Zhong et al., 2025), MMIN (Zhao et al., 2021), GCNet (Lian et al., 2023), IMDer (Wang et al., 2023), and MoMKE (Xu et al., 2024b). Table 4 reports the performance of UMQ under missing rates (MR) spanning 0.1 (mild) to 0.7 (severe). Across both datasets, UMQ surpasses strong baselines in nearly all missing configurations, confirming its robustness. **Averaged over all MRs, UMQ consistently ranks first**: on CMU-MOSI it yields 74.8% Acc2 and 37.8% Acc7, improving GCNet by 1.4 and 8.6 points; on CMU-MOSEI

Table 4: The results under missing modalities on the CMU-MOSI and CMU-MOSEI datasets.

|  | MR | CIDer
Acc2 / F1 / Acc7 | MMIN
Acc2 / F1 / Acc7 | GCNet
Acc2 / F1 / Acc7 | IMDer
Acc2 / F1 / Acc7 | MoMKE
Acc2 / F1 / Acc7 | UMQ
Acc2 / F1 / Acc7 |
|---|---|---|---|---|---|---|---|
| MOSI | 0.1 | 81.1 / 79.7 / 39.4 | 81.8 / 81.8 / 41.2 | 82.2 / 82.3 / 35.4 | 83.5 / 83.4 / 42.1 | 82.5 / 81.6 / 35.1 | **85.1 / 85.1 / 48.9** |
|  | 0.2 | 78.5 / 75.6 / 36.7 | 79.0 / 79.1 / 38.9 | 79.4 / 79.5 / 34.6 | 80.5 / 80.5 / 41.6 | 78.5 / 76.6 / 32.9 | **82.5 / 82.5 / 44.5** |
|  | 0.3 | 76.0 / 71.6 / 34.0 | 76.1 / 76.2 / 36.9 | 77.1 / 77.2 / 32.5 | 77.4 / 77.6 / 37.4 | 74.4 / 71.7 / 30.6 | **78.2 / 78.2 / 41.4** |
|  | 0.4 | 73.4 / 67.4 / 31.3 | 71.7 / 71.6 / 34.9 | 75.3 / 75.4 / 30.3 | 66.5 / 66.3 / 35.2 | 70.7 / 67.5 / 28.4 | **74.1 / 74.2 / 38.3** |
|  | 0.5 | 70.8 / 63.3 / 28.5 | 67.2 / 66.5 / **34.2** | **72.4 / 72.4** / 29.3 | 65.2 / 65.4 / 29.5 | 66.9 / 63.2 / 26.2 | 70.7 / 70.9 / 34.1 |
|  | 0.6 | 68.2 / 59.1 / 25.8 | 64.9 / 64.0 / 29.1 | 64.3 / 64.5 / 23.6 | 66.0 / 65.5 / 27.0 | 63.2 / 58.9 / 23.9 | **67.7 / 67.7 / 30.3** |
|  | 0.7 | 66.4 / 56.4 / 24.0 | 62.8 / 61.0 / **28.4** | 64.8 / 64.9 / 18.9 | 62.2 / 60.4 / 26.5 | 60.6 / 55.9 / 22.4 | **65.4 / 65.5** / 27.4 |
|  | Avg | 73.5 / 67.6 / 31.4 | 71.9 / 71.5 / 34.5 | 73.6 / 73.7 / 29.2 | 71.6 / 71.3 / 34.2 | 71.0 / 67.9 / 28.5 | **74.8 / 74.9 / 37.8** |
| MOSEI | 0.1 | 81.5 / 81.5 / 46.3 | 81.9 / 81.3 / 50.6 | 84.3 / 84.5 / 46.9 | 82.9 / 82.9 / 52.1 | 85.1 / 84.7 / 47.2 | **85.9 / 85.6 / 53.3** |
|  | 0.2 | 78.9 / 78.8 / 45.7 | 79.8 / 78.8 / 49.6 | 83.3 / 82.3 / 45.1 | 80.6 / 79.7 / 51.3 | 83.3 / 82.7 / 45.4 | **84.0 / 83.8 / 51.9** |
|  | 0.3 | 76.3 / 76.0 / 45.2 | 77.2 / 75.5 / 48.1 | 81.2 / 81.5 / 44.5 | 78.7 / 77.8 / **49.6** | 81.6 / 80.7 / 43.6 | **81.8 / 81.7** / 48.9 |
|  | 0.4 | 73.5 / 73.2 / 44.6 | 75.2 / 72.6 / 47.5 | 79.3 / 77.7 / 43.4 | 73.7 / 73.3 / 48.0 | 79.8 / 78.7 / 41.7 | **79.9 / 79.4 / 48.4** |
|  | 0.5 | 70.8 / 70.4 / 44.1 | 73.9 / 70.7 / 46.7 | 77.3 / 74.7 / 41.8 | 72.1 / 68.4 / 46.6 | **78.1** / 76.7 / 39.8 | 78.1 / **77.9 / 46.7** |
|  | 0.6 | 68.1 / 67.6 / 43.6 | 73.2 / 70.3 / 45.6 | 75.9 / 73.2 / 38.6 | 70.8 / 65.9 / 45.0 | **76.3** / 74.7 / 37.9 | 76.3 / **76.0 / 46.0** |
|  | 0.7 | 66.3 / 65.7 / 43.2 | 73.1 / 69.5 / 44.8 | 75.0 / 73.7 / 38.1 | 69.1 / 66.6 / 44.1 | **75.2** / 73.3 / 36.7 | 75.1 / **74.9 / 45.1** |
|  | Avg | 73.6 / 73.3 / 44.7 | 76.3 / 74.1 / 47.6 | 79.5 / 78.2 / 42.6 | 75.4 / 73.5 / 48.1 | 79.9 / 78.8 / 41.8 | **80.2 / 79.9 / 48.6** |

it reaches 80.2% Acc2 and 48.6% Acc7, outperforming MoMKE. These gains stem from UMQ's explicit simulation of modality absence via noise injection and quality enhancement that enables UMQ to cope with missing data and enhance low-quality features in incomplete settings.

## 4.4 DISCUSSION ON NOISY MODALITIES

For noisy modalities, we corrupt all modalities with Gaussian noise (noise rate 10%–70%) to assess the robustness of UMQ to modality noise. For comparison, we include C-MIB (Mai et al., 2023b) and Multimodal Boosting (Mai et al., 2024), two baselines that also address modality noise, and report Acc2 and MAE as per common practice. Table 5 shows that **UMQ outperforms all baselines across every noise level, most notably in MAE**, and maintains stable performance even when the noise rate reaches 0.7. This robustness arises from the design of our UMQ to comprehensively handle noisy input from quality estimation, quality enhancement to specialized modeling via MQ-MoE. In contrast, C-MIB only filters out noise and Multimodal Boosting only identifies noisy level. These findings underscore the superiority and robustness of UMQ in handling noise data. In section A.2, we show the results on other noises, verifying that **UMQ can generalize to unseen noises.**

Table 5: Discussion on noisy modalities.

|  | NR | C-MIB
Acc2/MAE | Multimodal Boosting
Acc2/MAE | UMQ
Acc2/MAE |
|---|---|---|---|---|
| MOSI | 0.1 | 87.8 / 0.670 | 86.7 / 0.678 | **88.2 / 0.627** |
|  | 0.2 | 87.5 / 0.726 | 86.1 / 0.738 | **87.8 / 0.652** |
|  | 0.3 | 86.4 / 0.912 | 86.4 / 0.785 | **87.6 / 0.644** |
|  | 0.4 | 83.2 / 1.366 | 85.5 / 0.841 | **86.9 / 0.643** |
|  | 0.5 | 84.9 / 1.660 | 86.1 / 1.172 | **88.1 / 0.673** |
|  | 0.6 | 80.8 / 2.595 | 82.0 / 1.355 | **86.9 / 0.763** |
|  | 0.7 | 82.1 / 3.146 | 84.4 / 1.750 | **85.8 / 0.764** |
|  | Avg | 84.7 / 1.582 | 85.3 / 1.046 | **87.3 / 0.681** |
| MOSEI | 0.1 | 86.1 / 0.545 | 86.4 / 0.544 | **87.3 / 0.531** |
|  | 0.2 | 84.5 / 0.582 | 86.6 / 0.557 | **87.0 / 0.528** |
|  | 0.3 | 85.6 / 0.622 | 85.5 / 0.623 | **86.7 / 0.530** |
|  | 0.4 | 84.4 / 0.703 | 85.3 / 0.682 | **87.0 / 0.536** |
|  | 0.5 | 83.7 / 0.875 | 84.1 / 0.724 | **86.6 / 0.543** |
|  | 0.6 | 82.4 / 1.054 | 85.4 / 0.924 | **85.6 / 0.573** |
|  | 0.7 | 80.5 / 1.404 | 80.3 / 1.125 | **85.5 / 0.616** |
|  | Avg | 83.9 / 0.826 | 84.8 / 0.740 | **86.5 / 0.551** |

## 4.5 ABLATION EXPERIMENTS

**(1) The importance of quality estimation**: As presented in Table 6, in the case of 'W/O Quality Estimation', we remove the quality estimator, and the components within MQ-MoE and quality enhancer that rely on quality scores are also removed. The model exhibits its sharpest performance decline, as quality estimation is not only the most pivotal component of UMQ but also the foundation upon which all other modules are based. In addition, the removal of the rank-guided training strategy causes a severe performance drop, as it constitutes the core technique for training the quality estimator and is the key to the accurate identification of the quality of modality.

Table 6: Ablation experiments.

| Model | CMU-MOSI | | |
|---|---|---|---|
|  | Acc7↑ | Acc2↑ | MAE↑ |
| W/O Quality Estimation | 44.8 | 86.4 | 0.676 |
| W/O Rank-Guided Training | 46.8 | 86.6 | 0.670 |
| W/O Quality Enhancement | 47.1 | 88.4 | 0.640 |
| W/O Modality Decoupling | 48.6 | 88.3 | 0.639 |
| W/O Modality-Specific Information | 48.1 | 88.1 | 0.641 |
| W/O Sample-Specific Information | 48.7 | 88.1 | 0.635 |
| W/O MQ-MOE | 48.4 | 87.6 | 0.639 |
| W/O $\mathcal{L}_{same}$ | 47.4 | 89.0 | 0.637 |
| UMQ | **49.7** | **90.1** | **0.630** |

**(2) The importance of quality enhancement**: Removing quality enhancement module also leads to a noticeable performance drop, because quality enhancer can strengthen modality representations. When any of its key sub-components (modality decoupling, modality-/sample-specific information)

is ablated, performance declines consistently. These results underscore the necessity of learning both modality-specific and sample-specific information for enhancing modality representations.

**(3) The importance of MQ-MoE**: In 'W/O MQ-MoE', model performance decreases by 2.5 points in Acc2, indicating MQ-MoE can improve the performance via modeling different types of input data in a more specialized manner. When $\mathcal{L}_{same}$ is removed, the performance drops by over 2 points in Acc7. This is mainly because $\mathcal{L}_{same}$ is the core constraint of MQ-MoE that ensures samples sharing an identical modality-quality configuration can be processed by the same set of experts, so that various inputs can be handled appropriately.

### 4.6 VISUALIZATION OF THE ENHANCED REPRESENTATIONS

To verify that modality-specific information is important to modality recovery, we present the visualization of the original and reconstructed language representations using t-SNE (Van der Maaten & Hinton, 2008) on the CMU-MOSI testing set. We first mix the original language features with Gaussian noise to generate corrupted features, and then use the quality enhancer to generate two versions of enhanced features (with and without modality baseline representation $x_m^b$). As shown in Figure 3, features enhanced by the quality enhancer in our UMQ lie closer to the original features in the feature space and exhibit partial overlap with the original ones, whereas features generated without modality-specific information are markedly more distant and display virtually no overlapping regions. The visualization suggests that using modality baseline representation to enhance low-quality data can produce more similar features to the original ones.

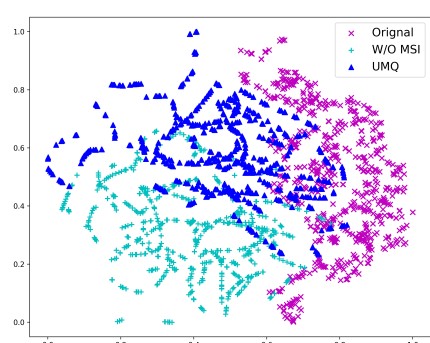

Figure 3: Visualization of the original and reconstructed language features. 'UMQ' and 'W/O MSI' denote the reconstructed features obtained by our method and obtained without modality-specific information, respectively.

### 4.7 CASE STUDY ON ESTIMATED QUALITY AND EXPERTS

We use three examples to illustrate the effectiveness of the quality estimator and MQ-MoE (we construct 10 experts and each sample is routed to 3 experts.). As shown in Figure 4, the quality estimator accurately assigns a very low score to noisy modalities. Meanwhile, the comparatively high rating for the acoustic modality of sample 1 compared to other samples reveals that the estimator can identify more discriminative information and assign appropriate scores. Moreover, it can be observed that MQ-MoE correctly assigns same experts to two samples (samples 1 and 2) sharing an identical modality-quality configuration, while allocating distinct experts to samples with differing configurations, demonstrating the effectiveness of the proposed routing strategy.

Figure 4: Analysis on estimated quality and selected experts. The modalities enclosed by the red bounding box are replaced with Gaussian noise.

## 5 CONCLUSION

We jointly handle missing and noisy modalities to enhance model robustness for low-quality data. UMQ trains the quality estimator with explicit supervised signals via a rank-guided training strategy, and uses sample- and modality-specific information to enhance the quality of modality representations. MQ-MoE is then constructed to enable multiple modality-quality problems to be addressed specifically. UMQ outperforms competitive baselines in various settings.

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

# Appendix

CONTENTS

Table 8: Complete results on CMU-MOSI and CMU-MOSEI under complete modalities. The results labeled with † are obtained from original papers, and other results are obtained from our experiments. The best results are highlighted in bold and the second best results are marked with underlines.

| | CMU-MOSI | | | | | CMU-MOSEI | | | | |
|---|---|---|---|---|---|---|---|---|---|---|
| | Acc7↑ | Acc2↑ | F1↑ | MAE↓ | Corr↑ | Acc7↑ | Acc2↑ | F1↑ | MAE↓ | Corr↑ |
| MFM (Tsai et al., 2019) | 33.3 | 80.0 | 80.1 | 0.948 | 0.664 | 50.8 | 83.4 | 83.4 | 0.580 | 0.722 |
| Self-MM (Yu et al., 2021) | 45.8 | 84.9 | 84.8 | 0.731 | 0.785 | 53.0 | 85.2 | 85.2 | 0.540 | 0.763 |
| AtCAF† (Huang et al., 2025) | 46.5 | 88.6 | 88.5 | 0.650 | 0.831 | 55.9 | 87.0 | 86.8 | 0.508 | 0.785 |
| DLF† (Wang et al., 2025) | 47.1 | 85.1 | 85.0 | 0.731 | 0.781 | 53.9 | 85.4 | 85.3 | 0.536 | 0.764 |
| DEVA† (Wu et al., 2025) | 46.3 | 86.3 | 86.3 | 0.730 | 0.787 | 52.3 | 86.1 | 86.2 | 0.541 | 0.769 |
| C-MIB (Mai et al., 2023b) | 47.7 | 87.8 | 87.8 | 0.662 | 0.835 | 52.7 | 86.9 | 86.8 | 0.542 | 0.784 |
| ITHP (Xiao et al., 2024) | 47.7 | 88.5 | 88.5 | 0.663 | 0.856 | 52.2 | 87.1 | 87.1 | 0.550 | 0.792 |
| Multimodal Boosting (Mai et al., 2024) | 49.1 | 88.5 | 88.4 | 0.634 | 0.855 | 54.0 | 86.5 | 86.5 | 0.523 | 0.779 |
| HumanOmni (Zhao et al., 2025) | - | 76.9 | 76.9 | - | - | - | 80.5 | 80.4 | - | - |
| Qwen2.5Omni (Xu et al., 2025) | - | 84.4 | 84.3 | - | - | - | 84.0 | 83.2 | - | - |
| MiniCPM-o (Yao et al., 2024) | - | 80.0 | 80.1 | - | - | - | 77.4 | 77.3 | - | - |
| VideoLLaMA2-AV (Cheng et al., 2024) | - | 80.2 | 79.6 | - | - | - | 76.8 | 77.2 | - | - |
| UMQ | 49.7 | 90.1 | 90.0 | 0.630 | 0.863 | 55.5 | 88.1 | 88.1 | 0.506 | 0.796 |

# A  ADDITIONAL EXPERIMENTAL RESULTS

## A.1  ADDITIONAL RESULTS ON COMPLETE MODALITIES

(1) **Results on the CH-SIMS dataset**:

On the CH-SIMS dataset, UMQ leads in all evaluation metrics except for the MAE metric, where it ranks second, marginally behind ConFEDE (Yang et al., 2023). Overall, considering the results across the three widely-used datasets, UMQ demonstrates state-of-the-art performance in the task of MSA, further verifying the effectiveness of the proposed unified modality-quality framework.

Table 7: The results on CH-SIMS dataset. The results labeled with † are obtain in their original paper.

| | MAE ↓ | Corr ↑ | Acc2 ↑ | F1-score ↑ |
|---|---|---|---|---|
| MFM (Tsai et al., 2019) | 0.493 | 0.473 | 71.7 | 70.5 |
| Self-MM (Yu et al., 2021) | 0.425 | 0.592 | 80.0 | 80.4 |
| ConFEDE (Yang et al., 2023) | 0.394 | 0.637 | 81.0 | 81.0 |
| DEVA† (Wu et al., 2025) | 0.424 | 0.583 | 79.6 | 80.3 |
| ITHP (Xiao et al., 2024) | 0.425 | 0.581 | 79.4 | 79.4 |
| HumanOmni (Zhao et al., 2025) | - | - | 79.4 | 80.2 |
| Qwen2.5Omni (Xu et al., 2025) | - | - | 82.5 | 81.6 |
| MiniCPM-o (Yao et al., 2024) | - | - | 76.2 | 73.4 |
| VideoLLaMA2-AV (Cheng et al., 2024) | - | - | 76.8 | 77.2 |
| UMQ | 0.401 | 0.668 | 82.9 | 82.6 |

(2) **Comparison with multimodal large language models (MLLMs)**:

Owing to the robust capacity of MLLMs to manage diverse modalities, we evaluate the efficacy of MLLMs that are capable of simultaneously processing visual, acoustic, and textual inputs. This includes models such as HumanOmni (Zhao et al., 2025), Qwen2.5Omni (Xu et al., 2025), MiniCPM-o (Yao et al., 2024), and VideoLLaMA2-AV (Cheng et al., 2024). Given that MLLMs often face challenges in precisely determining continous sentiment scores, our assessment is limited to binary sentiment classification tasks, where the models are required to classify a sample's sentiment as either positive or negative. For those MLLMs that fail to produce positive/negative indicators, we ascertain the token ids associated with these sentiments and make predictions based on which token id is assigned a higher likelihood. As detailed in Tables 8 and 7, MLLMs demonstrate commendable performance, outperforming several baseline models that have undergone dataset-specific fine-tuning. This suggests that MLLMs have an innate aptitude for sentiment analysis tasks. Among the MLLMs, Qwen2.5Omni stands out as the top performer and even secures the second-highest scores on the CH-SIMS dataset. Nonetheless, it still falls short when compared to the proposed UMQ, indicating that MLLMs need additional dataset-specific fine-tuning to enhance their performance.

## A.2  ADDITIONAL RESULTS ON OTHER TYPES OF NOISE

In addition to Gaussian noise, we have evaluated the robustness of UMQ against Laplace noise and random erasing noise (randomly select a portion of the features and set them to zero to simulate data loss), which are not used in the 'AddNoise' function for training. For each sample, we randomly select one type of noise to add to the features, and the results are shown in Table 9. The noisy rate (NR) is set to 10% - 70%. For a comprehensive and fair comparison, we present the results of NIAT (Yuan et al., 2024), C-MIB (Mai et al., 2023b), and Multimodal Boosting (Mai et al., 2024) in Table 9, which use the same training and testing settings as our UMQ (we also present the complete

Table 9: Additional results on other types of noise on the CMU-MOSI and CMU-MOSEI datasets.

| | NR | NIAT
Acc2 / MAE | C-MIB
Acc2 / MAE | Multimodal Boosting
Acc2 / MAE | UMQ
Acc2 / MAE |
|---|---|---|---|---|---|
| MOSI | 0.1 | 85.5 / **0.622** | 87.6 / 0.681 | 87.3/0.660 | **88.3** / 0.625 |
| | 0.2 | 84.1 / 0.684 | 87.3 / 0.695 | 85.8/0.726 | **87.4 / 0.659** |
| | 0.3 | 83.4 / 0.777 | 85.0 / 0.890 | 85.6/0.757 | **87.6 / 0.655** |
| | 0.4 | 82.2 / 0.825 | 83.7 / 1.019 | 87.6/0.798 | **87.9 / 0.679** |
| | 0.5 | 80.5 / 1.031 | 81.5 / 1.629 | 86.9/0.839 | **87.6 / 0.656** |
| | 0.6 | 79.7 / 1.115 | 79.4 / 1.274 | 84.6/1.048 | **87.5 / 0.706** |
| | 0.7 | 79.1 / 1.237 | 81.4 / 3.443 | 85.4/1.432 | **85.8 / 0.732** |
| | **Avg** | 82.1 / 0.899 | 83.7 / 1.376 | 86.2/0.894 | **87.4 / 0.673** |
| MOSEI | 0.1 | 83.9 / 0.557 | 86.8 / 0.531 | 86.8 / 0.555 | **87.2 / 0.525** |
| | 0.2 | 81.6 / 0.573 | 85.9 / 0.587 | 86.0 / 0.581 | **87.0 / 0.523** |
| | 0.3 | 81.0 / 0.612 | 85.7 / 0.606 | 85.2 / 0.586 | **87.1 / 0.529** |
| | 0.4 | 79.1 / 0.651 | 84.0 / 0.646 | 85.4 / 0.680 | **86.7 / 0.527** |
| | 0.5 | 77.9 / 0.716 | 83.9 / 0.798 | 85.0 / 0.746 | **86.6 / 0.541** |
| | 0.6 | 77.5 / 0.740 | 82.2 / 1.046 | 86.0 / 0.957 | **86.3 / 0.562** |
| | 0.7 | 76.1 / 0.802 | 77.6 / 1.271 | 81.2 / 1.018 | **84.4 / 0.584** |
| | **Avg** | 79.6 / 0.664 | 83.7 / 0.784 | 85.1 / 0.732 | **86.5 / 0.542** |

Table 10: Complete Results on noisy modalities (Gaussian noise) on the CMU-MOSI and CMU-MOSEI datasets.

| | NR | NIAT
Acc2/MAE | C-MIB
Acc2/MAE | Multimodal Boosting
Acc2/MAE | UMQ
Acc2/MAE |
|---|---|---|---|---|---|
| MOSI | 0.1 | 83.6 / 0.672 | 87.8 / 0.670 | 86.7 / 0.678 | **88.2 / 0.627** |
| | 0.2 | 83.0 / 0.689 | 87.5 / 0.726 | 86.1 / 0.738 | **87.8 / 0.652** |
| | 0.3 | 82.3 / 0.741 | 86.4 / 0.912 | 86.4 / 0.785 | **87.6 / 0.644** |
| | 0.4 | 81.7 / 0.847 | 83.2 / 1.366 | 85.5 / 0.841 | **86.9 / 0.643** |
| | 0.5 | 80.2 / 0.949 | 84.9 / 1.660 | 86.1 / 1.172 | **88.1 / 0.673** |
| | 0.6 | 78.9 / 1.153 | 80.8 / 2.595 | 82.0 / 1.355 | **86.9 / 0.763** |
| | 0.7 | 77.6 / 1.290 | 82.1 / 3.146 | 84.4 / 1.750 | **85.8 / 0.764** |
| | **Avg** | 81.0 / 0.910 | 84.7 / 1.582 | 85.3 / 1.046 | **87.3 / 0.681** |
| MOSEI | 0.1 | 80.3 / 0.554 | 86.1 / 0.545 | 86.4 / 0.544 | **87.3 / 0.531** |
| | 0.2 | 79.1 / 0.593 | 84.5 / 0.582 | 86.6 / 0.557 | **87.0 / 0.528** |
| | 0.3 | 77.7 / 0.637 | 85.6 / 0.622 | 85.5 / 0.623 | **86.7 / 0.530** |
| | 0.4 | 77.2 / 0.672 | 84.4 / 0.703 | 85.3 / 0.682 | **87.0 / 0.536** |
| | 0.5 | 76.3 / 0.735 | 83.7 / 0.875 | 84.1 / 0.724 | **86.6 / 0.543** |
| | 0.6 | 74.9 / 0.797 | 82.4 / 1.054 | 85.4 / 0.924 | **85.6 / 0.573** |
| | 0.7 | 73.4 / 0.863 | 80.5 / 1.404 | 80.3 / 1.125 | **85.5 / 0.616** |
| | **Avg** | 77.0 / 0.673 | 83.9 / 0.826 | 84.8 / 0.740 | **86.5 / 0.551** |

results of Gaussian noise in Table 10 for completeness). We reproduce the results of the baselines using the same training and testing settings as our UMQ to make a fair comparison. It can be seen from Table 9 that our UMQ still significantly outperforms competitive baselines under other noises that are unseen during training (especially when the noise rate is large), further demonstrating the effectiveness and generalization ability of UMQ.

## A.3 HYPERPARAMETER ROBUSTNESS ANALYSIS

In this section, we evaluate the effectiveness of hyperparameters on the CMU-MOSEI dataset, including the weight for modality decoupling loss $\beta_{de}$, the weight for MQ-MoE loss $\beta_{moe}$, the weight for quality estimator loss $\beta_{est}$, the number of selected experts $k$, and the number of total experts $h$.

The results with the loss weights $\beta_{de}$, $\beta_{moe}$, and $\beta_{est}$ setting to different values are presented in Fig. 5 (a), (b), and (c), respectively. For all loss weights, the model performs best when they are set to moderate values, and there is a noticeable decline in performance when their values are either too large or too small. This is mainly because when the weights are too small, the effect of the designed losses is not significant enough, and when they are too large, the designed auxiliary losses can interfere with the learning of the main task loss. We have found in our experiments that the

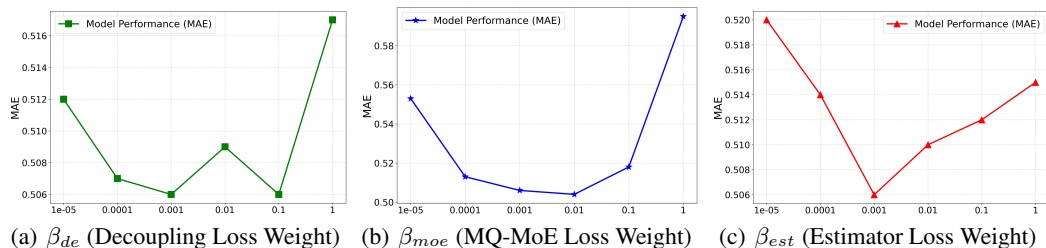

(a) $\beta_{de}$ (Decoupling Loss Weight)  (b) $\beta_{moe}$ (MQ-MoE Loss Weight)  (c) $\beta_{est}$ (Estimator Loss Weight)

Figure 5: The MAE of UMQ with respect to the change of loss weights on CMU-MOSEI.

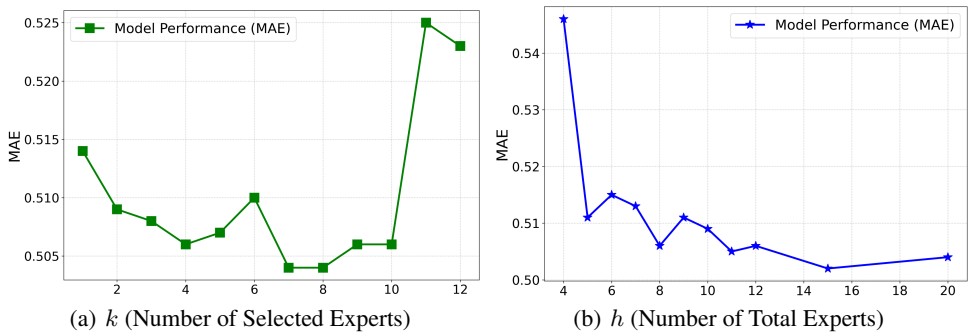

(a) $k$ (Number of Selected Experts)  (b) $h$ (Number of Total Experts)

Figure 6: The MAE of UMQ with respect to the change of the number of total experts $h$ and the number of selected experts $k$ on CMU-MOSEI.

model performance with respect to the change of other loss weights is similar, so we only present the results of the above losses.

The results with the number of selected experts $k$ and the number of total experts $h$ setting to different values are presented in Fig. 6 (a) and (b), respectively. As shown in Fig. 6 (a), When the number of selected experts $k$ is set to 7 or 8, the model achieves the best performance, surpassing the default setting ($k = 4$), and there is a noticeable decline in performance when $k$ is greater than 10. This is because when $k$ exceeds 10, the number of experts selected for each sample is very close to or equal to the total number of experts ($h = 12$), at which point nearly all samples share the same set of experts. Samples with different modality quality configurations are processed in the same way, making it difficult to achieve more targeted processing for each specific modality quality configuration. Conversely, when $k$ is too small, the number of experts handling each sample is insufficient, leading to a decrease in expressive power, and thus the performance of UMQ is not as good as when $k$ is of moderate size. As for the number of total experts $h$, it can be observed from Fig. 6 (b) that as the value of $h$ increases, the performance of UMQ generally improves. This is because, with a larger $h$, there are more experts available to choose from, and the expressive capability of the model is enhanced. For the sake of model complexity, we do not infinitely increase $h$, but instead select it within a reasonable range and set the default value of $h$ to 12 after a random search of fifty-times.

In particular, we find that when the hyperparameters are set to some specific values (e.g., when $k$ is 7), the performance of UMQ is better than that obtained by random grid search, indicating that the performance of UMQ can be further improved after a more careful hyperparameter tuning and suggesting the potential of UMQ.

## B IMPLEMENTATION DETAILS

### B.1 UNIMODAL NETWORKS

In this part, we describe the architectures of unimodal networks and demonstrate the generation of unimodal representations for subsequent integration. To be consistent with state-of-the-art techniques (Xiao et al., 2024; Hasan et al., 2021; Wang et al., 2025), we employ pre-trained language models (PLMs) (He et al., 2021; Lan et al., 2020) to produce advanced textual features. The proce-

dure of the language network for all downstream tasks is outlined as follows.:

$$\hat{\boldsymbol{U}}_l = \text{PLM}(\boldsymbol{U}_l;\ \theta_l) \in \mathbb{R}^{T_l \times d_l}$$
$$\boldsymbol{X}_l = (\hat{\boldsymbol{U}}_l \boldsymbol{W}_{pro} + \boldsymbol{b}_{pro}) \in \mathbb{R}^{T_l \times d} \tag{24}$$

where $\boldsymbol{U}_l$ denotes the input token sequence and $T_l$ is the length of the token sequence. $\boldsymbol{W}_{pro} \in \mathbb{R}^{d_l \times d}$ and $\boldsymbol{b}_{pro} \in \mathbb{R}^{1 \times d}$ are trainable parameters that transform the output dimensionality of the language network to the shared feature dimensionality $d$. For the MSA task, the operational procedures for both the acoustic and visual networks, which utilize transformer encoders (Vaswani et al., 2017), are presented below ($m \in \{a, v\}$):

$$\hat{\boldsymbol{U}}_m = \text{Conv 1D}(\boldsymbol{U}_m;\ K_m) \in \mathbb{R}^{T_m \times d}$$
$$\boldsymbol{X}_m = \text{Transformer}(\hat{\boldsymbol{U}}_m;\ \theta_m) \in \mathbb{R}^{T_m \times d} \tag{25}$$

where $\text{Conv 1D}$ represents the temporal convolution with kernel size $K_m$ being 3.

It should be noted that, for the MHD and MSD tasks, in order to effectively detect humor-related content, we follow previous approaches (Hasan et al., 2021; Mai et al., 2023c) by extracting Human Centric Features (HCF) from the language modality to complement as an additional fourth modality (see (Hasan et al., 2021) for more details) and is represented as $\boldsymbol{U}_h \in \mathbb{R}^{T_l \times d_h}$. Moreover, each instance comprises a target punchline segment accompanied by its preceding contextual segments. By merging the feature sequences of the punchline and the context segments along the temporal axis, we generate the unimodal inputs $\boldsymbol{U}_m \in \mathbb{R}^{T_m \times d_m}$ ($m \in \mathcal{M} = \{a, v, l, h\}$). The unimodal network designed for the HCF modality, which is also composed of transformer encoders, shares a comparable structure with those utilized for visual and acoustic modalities. Specifically, for the MHD and MSD tasks, the operational procedures of the transformer-based unimodal networks are detailed as follows ($m \in \{a, v, h\}$):

$$\hat{\boldsymbol{U}}_m = \text{Transformer}(\boldsymbol{U}_m;\ \theta_m) \in \mathbb{R}^{T_m \times d_m}$$
$$\boldsymbol{X}_m = \text{Conv 1D}(\hat{\boldsymbol{U}}_m;\ K_m) \in \mathbb{R}^{T_m \times d} \tag{26}$$

Furthermore, to streamline the subsequent modeling procedure, we combine the language and HCF modalities using a simple linear layer:

$$\boldsymbol{X}_l \longleftarrow \text{Linear}(\boldsymbol{X}_l \oplus \boldsymbol{X}_h;\ \theta_{lin}) \in \mathbb{R}^{T_l \times d} \tag{27}$$

For all tasks, given the extracted unimodal representation $\boldsymbol{X}_m \in \mathbb{R}^{T_m \times d}$, we use a layer normalization to regularize the feature distributions, and then conduct mean pooling at the temporal dimension to obtain a feature vector that simplifies the later processing by the quality estimator:

$$\boldsymbol{x}_m = MeanPooling(LN(\boldsymbol{X}_m)) \in \mathbb{R}^{1 \times d} \tag{28}$$

The obtained $\boldsymbol{x}_m$ is used for quality estimation.

## B.2    DATASET COMPOSITION

We apply the following widely-used datasets to evaluate the effectiveness of UMQ:

(1) **CMU-MOSI** (Zadeh et al., 2016): CMU-MOSI is a dataset extensively used for MSA, comprising over 2,000 video segments collected from the Internet. Each segment is manually assigned a sentiment score ranging from -3 to 3, with 3 indicating the most intense positive sentiment and -3 indicating the most intense negative sentiment.

(2) **CMU-MOSEI** (Zadeh et al., 2018): CMU-MOSEI represents a large-scale MSA dataset, encompassing more than 22,000 video segments from over 1,000 YouTube speakers on 250 diverse subjects. The segments are randomly chosen from an assortment of topics and individual video presentations. Each video segment is marked with two types of labels: emotions categorized into six distinct classes and sentiment scores that vary from -3 to 3. To evaluate UMQ's performance on the MSA task, we utilize the sentiment labels from the CMU-MOSEI dataset, which correspond to the sentiment scale of the CMU-MOSI dataset.

(3) **CH-SIMS** (Yu et al., 2020) dataset is distinguished as a dedicated resource for multimodal sentiment analysis in Chinese, consisting of 2,281 premium video segments sourced from films, television series, and chat programs. These segments exhibit a spectrum of spontaneous emotional displays under varying head positions, obstructions, and illumination settings. Human evaluators have carefully assigned a sentiment rating to each segment, ranging from -1, indicating extremely negative, to 1, indicating extremely positive. Following the partitioning approach of prior research (Yu et al., 2021; Han et al., 2021), the dataset is organized into 1,368 utterances for the training phase, 456 for the validation phase, and 457 for the testing phase.

(4) **UR-FUNNY** (Hasan et al., 2019): Created for the MHD task, the UR-FUNNY dataset consists of TED talk videos from 1,741 speakers. In this dataset, each target video segment, labeled as a 'punchline', includes language, acoustic, and visual information. The video segments that come before these punchlines, called context segments, are also fed into the model to provide context for analysis. Punchlines are recognized by the 'laughter' marker in the transcripts, marking the times when the audience laughed during the presentation. Negative examples are identified using a comparable approach, where the target punchline segments are not accompanied by the 'laughter' marker. The UR-FUNNY dataset includes 7,614 samples for training, 980 for validation, and 994 for testing. To maintain consistency with the latest research methods (Hasan et al., 2021; Mai et al., 2023a;c), we use version 2 of the UR-FUNNY dataset to assess our proposed model.

(5) **MUStARD** (Castro et al., 2019): MUStARD is a dataset for sarcasm detection that originates from several popular TV series, such as Friends, The Big Bang Theory, The Golden Girls, and Sarcasmolics. It comprises 690 video segments, each of which has been manually classified as either sarcastic or non-sarcastic. Beyond the punchline segments, MUStARD also includes the preceding dialogues (context segments) for each punchline to offer contextual details.

## B.3 ASSESSMENT CRITERIA

For assessing the MSA task, we measure the performance of UMQ and the baselines based on the following criteria: (1) **Acc7**: This metric evaluates how well the model can categorize sentiment scores into seven distinct classes. To compute Acc7, both predictions and labels are rounded to the nearest integer value between -3 and 3; (2) **Acc2**: This metric assesses the model's capability to correctly differentiate between positive and negative sentiments in a binary classification scenario; (3) **F1 score**: This is a metric that combines precision and recall for binary sentiment classification tasks. When calculating both Acc2 and the F1 score, neutral segments are ignored; (4) **MAE**: This represents the mean absolute error between the model's predictions and the actual labels; (5) **Corr**: This is the correlation coefficient that measures the strength and direction of the relationship between the model's predictions and the actual labels.

For the MHD and MSD tasks, consistent with previous methodologies (Hasan et al., 2021; Mai et al., 2023a;c), we present the model's binary accuracy (i.e., whether it classifies samples as humorous or not, sarcastic or not).

## B.4 FEATURE EXTRACTION STRATEGY

We applies the following methods to extract the features of each modality:

**(1) Visual Modality**: In line with previous studies (Xiao et al., 2024), for the CMU-MOSI and CMU-MOSEI datasets, we employ Facet[1] to extract visual features including facial action units, facial landmarks, and head positioning, creating a temporal sequence that captures facial expressions and body gestures over time. For the CH-SIMS dataset, following previous works (Yu et al., 2021; 2020), we use the MTCNN face detection algorithm (Zhang et al., 2016) to obtain aligned facial images. Following this, the MultiComp OpenFace2.0 toolkit (Baltrusaitis et al., 2018) is applied to extract facial features, including 68 facial landmarks, 17 facial action units, as well as head pose, orientation, and eye gaze data. For the MHD and MSD tasks, aligning with current state-of-the-art approaches (Hasan et al., 2021; Mai et al., 2023c), we utilize OpenFace 2 (Baltrusaitis et al., 2018) to extract facial action units, as well as rigid and non-rigid facial shape parameters, among others.

---

[1]iMotions 2017. https://imotions.com/

**(2) Acoustic Modality**: For the CH-SIMS dataset, following prior methods (Yu et al., 2021; 2020), we employ the LibROSA (McFee et al., 2015) speech toolkit with its default settings to extract acoustic features at a frequency of 22050Hz. The extracted 33-dimensional frame-level features include 1-dimensional log F0, 20-dimensional MFCCs, and 12-dimensional CQT. For other datasets, we employ COVAREP (Degottex et al., 2014) for the extraction of time-series acoustic features from audio segments, encompassing 12 Mel-frequency cepstral coefficients, pitch tracking, speech polarity, glottal closure moments, spectral envelope, and more. These features, extracted across the entirety of each audio segment, capture the dynamic fluctuations in vocal tone throughout the speech.

**(3) Language Modality**: In the CMU-MOSI and CMU-MOSEI datasets, adhering to state-of-the-art approaches (Xiao et al., 2024), we utilize DeBERTa (He et al., 2021) to obtain advanced textual features. For the CH-SIMS dataset, to make a fair comparison with previous methods (Xie et al., 2024; Yu et al., 2021), BERT-cn (Devlin et al., 2019) is used as the language network. For the MHD and MSD tasks, in line with contemporary methodologies (Hasan et al., 2021; Mai et al., 2023a), ALBERT (Lan et al., 2020) is implemented as the language model. Notably, for MHD and MSD, we combine the sequences of punchlines and context tokens to produce the ultimate input for the language model: $U_l = C_l \oplus [SEP] \oplus P_l$, where the $[SEP]$ token serves to distinguish between the context token $C_l$ and the punchline tokens $P_l$ (Hasan et al., 2021).

For the CMU-MOSI dataset, the dimensions of the language, acoustic, and visual features are 768, 74, and 47, respectively. As for the CMU-MOSEI dataset, the dimensions of the corresponding uni-modal features are 768, 74, and 35, respectively. For the CH-SIMS dataset, the language, acoustic, and visual modalities have input dimensionalities of 768, 33, and 709, respectively. Regarding the UR-FUNNY and MUStARD datasets, the dimensions of the language, acoustic, visual, and HCF features are 768, 60, 36, and 4, respectively. For detailed information on the feature extraction process for the HCF modality, please see (Hasan et al., 2021).

### B.5 EXPERIMENTAL DETAILS

(1) **Hyperparameter setting**: We develop the proposed UMQ using the PyTorch framework on an NVIDIA RTX2080Ti GPU, equipped with CUDA version 11.6 and PyTorch version 1.13.1, and employ the AdamW optimizer (Loshchilov & Hutter, 2019) to train the model. The hyperparameter configurations are specified in Table 11. To be consistent with prior research (Gkoumas et al., 2021), we establish a feasible search range for each hyperparameter and conduct a random grid search with 50 iterations on the validation set to identify the most effective hyperparameters. We retain the top-performing hyperparameter configuration identified during the random search (based on the loss metric). After hyperparameter searching, we retrain the model for five times with the optimal settings under various random seeds, and present the average outcomes from these five runs.

(2) **Structures of the components**: The structures of the modality decouple network, modality couple network, fusion network and predictor are shown in Figure 7, and the structures of quality estimator, quality enhancer, and experts are presented in Figure 8. Notably, in the quality enhancer, we define a learnable query $q_m \in \mathbb{R}^{1 \times d}$ to aggregate the information within sample-specific information $\{x_{m'}^s \cdot \alpha_{m'} | m' \neq m\}$ and modality-specific information $x_m^b$, and the resulted representation is concatenated with $x_m$ to explicitly provide modality- and sample-specific information.

(3) **Protocol for the evaluation of missing modalities**: For assessing the impact of absent modalities, we conduct an extensive evaluation of the effectiveness of different methods on multimodal datasets with differing levels of modality missing. The rate of missing modality is determined as:

$$MR = 1 - \frac{\sum_{i=1}^{N} M_i}{N \times |\mathcal{M}|} \qquad (29)$$

where $M_i$ denotes the number of available modalities in the $i$-th sample, $N$ is the total number of samples, and $|\mathcal{M}|$ is the number of modalities. For every sample possessing $|\mathcal{M}|$ modalities, we randomly mask a subset of these modalities with a probability that matches the specified missing rate $MR$, while guaranteeing that for each sample at least one modality remains accessible, i.e., $M_i \geq 1$. This constraint guarantees that the missing rate does not exceed $\frac{|\mathcal{M}|-1}{|\mathcal{M}|}$. In our experiments, the number of modalities is $|\mathcal{M}| = 3$, and we choose $MR$ from the set $\{0.1, 0.2, \ldots, 0.7\}$. Particularly, when $MR = 0.7$, each sample randomly preserves one modality, epresenting the most

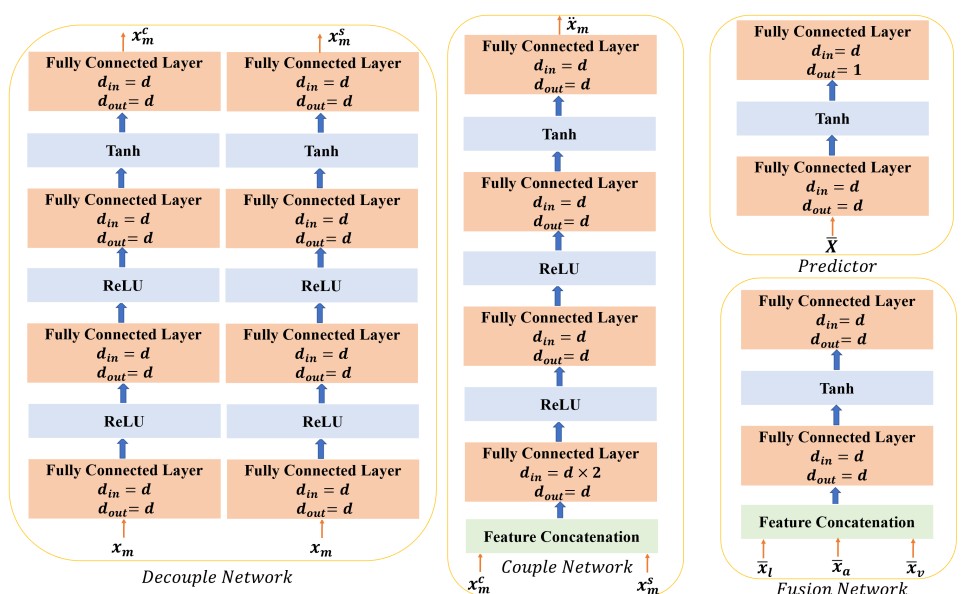

Figure 7: The structures of modality decouple network, modality couple network, fusion network and predictor.

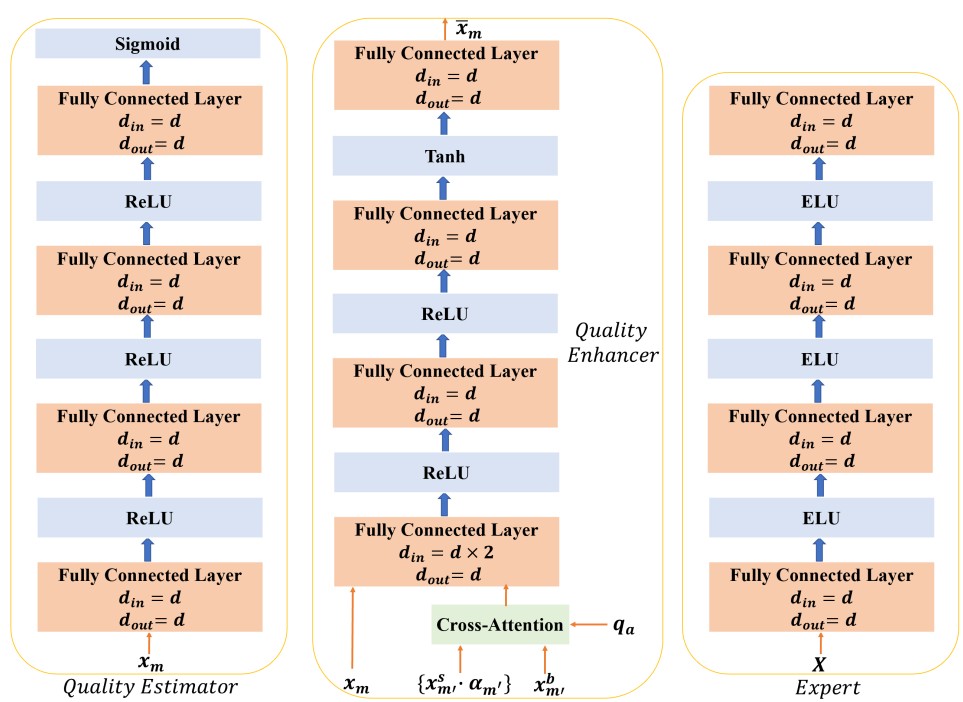

Figure 8: The structures of quality estimator, quality enhancer, and experts.

Table 11: **Hyperparameter Settings of UMQ.** MSE and BCE denotes mean square error and binary cross-entropy, respectively.

| | CMU-MOSI | CMU-MOSEI | CH-SIMS | UR-FUNNY | MUStARD |
|---|---|---|---|---|---|
| Loss Function | MSE | MSE | MSE | BCE | BCE |
| Batch Size | 64 | 64 | 32 | 48 | 48 |
| Learning Rate | 1e-5 | 1e-5 | 1e-5 | 5e-6 | 5e-6 |
| Shared Dimensionality $d$ | 100 | 100 | 100 | 100 | 64 |
| Variance Margin $\beta$ | 0.1 | 0.1 | 0.1 | 0.1 | 0.1 |
| Margin $\gamma_1$ | 0.05 | 0.1 | 0.1 | 0.1 | 0.1 |
| Maximum distance $\epsilon$ | 0.2 | 0.2 | 0.1 | 0.1 | 0.2 |
| Quality Threshold $\tau$ | 0.5 | 0.6 | 0.5 | 0.5 | 0.6 |
| Number of Experts $h$ | 10 | 12 | 12 | 10 | 10 |
| Selected Experts $k$ | 3 | 4 | 3 | 3 | 3 |
| Decouple Loss Weight $\beta_{de}$ | 1e-5 | 0.001 | 1e-4 | 1e-4 | 0.001 |
| Estimator Loss Weight $\beta_{est}$ | 0.001 | 0.001 | 0.001 | 1e-4 | 0.005 |
| Enhancer Loss Weight $\beta_{en}$ | 0.001 | 0.001 | 0.005 | 1e-4 | 0,001 |
| MQ-MoE Loss Weight $\beta_{est}$ | 0.001 | 0.001 | 0.005 | 0.1 | 1e-4 |

extreme case of modality missing. To make a fair comparison with our baselines, the same missing rate configuration is applied across the training, validation, and test phases, following the protocol adopted in previous works (Wang et al., 2023; Lian et al., 2023). Notably, for the experiments regarding missing modalities, to make a fair comparison with our baselines, we use the same feature sets as our closest baselines (Xu et al., 2024b; Lian et al., 2023).

(4) **Protocol for the evaluation of noisy modalities**: For the evaluation of noisy modalities, we generate the noisy features using the following equation:

$$\boldsymbol{x}_m^n = (1 - NR) \cdot \boldsymbol{x}_m + NR \cdot \mathcal{N} \tag{30}$$

where $\boldsymbol{x}_m$ is the unimodal representation, $NR$ is the noisy rate ranging from 0.1 to 0.7, $\mathcal{N}$ is the Gaussian noise data of mean 0 and variance 1, and $\boldsymbol{x}_m^n$ is the noisy representation that is used for the later quality estimator. Importantly, we apply the above noise mixing operation to all modalities of every sample in order to simulate real-world noise. Given that input-level noise ultimately impacts the features, it is reasonable to introduce noise at the feature level. Furthermore, to ensure fairness and protect privacy, multimodal affective computing algorithms usually model using the same set of features. This makes it even more practical to apply noise at the feature level. We reproduce the results of the baselines (Yuan et al., 2024; Mai et al., 2023b; 2024) using the same training and testing settings as our UMQ to make a fair comparison.

Additionally, to test the robustness of our model to other types of noise (see Table 9), we further evaluate the robustness of UMQ against Laplace noise and random erasing noise (randomly select a portion of the features and set them to zero to simulate data loss). At this point, for each sample, there is a 50% probability that the features will be mixed with Laplacian noise using Equation 30, and a 50% probability that the features will experience random feature dropout (random feature missing), where the dropout ratio is $NR$.

## B.6 BASELINES

The selected baselines for the MSA task include:

(1) **Multimodal Factorization Model** (**MFM**) (Tsai et al., 2019): MFM separates down multimodal features into two separate sets of factors: multimodal discriminative factors and modality-specific generative factors. This separation aids in the learning of effective multimodal representations, and the resulting factorized features can be utilized to comprehend essential inter-modal interactions in multimodal learning. (2) **Information-Theoretic Hierarchical Perception** (**ITHP**) (Xiao et al., 2024): Adhering to the information bottleneck principle, ITHP identifies a single core modality and treats the rest of the modalities as detectors within the information pathway, whose role is to refine the information flow. (3) **Multimodal Boosting** (Mai et al., 2024): It utilizes several base learners, each of which concentrates on distinct facets of multimodal learning. To evaluate their individual contributions, Multimodal Boosting incorporates a contribution learning module that dynamically assesses each base learner's contribution and the noise level of unimodal representations. (4) **Complete Multimodal Information Bottleneck** (**C-MIB**) (Mai et al., 2023b): It employs the

information bottleneck principle to diminish redundancy and noise within both unimodal and multimodal features, serving as the baseline for handling noisy modalities. (5) **Self-Supervised Multi-task Multimodal sentiment analysis network** (**Self-MM**) (Yu et al., 2021): It derives sentiment labels for individual modalities by utilizing the global labels of multimodal samples through a self-supervised approach, which in turn enables the extraction of more discriminative unimodal features. (6) **Attention-based Causality-Aware Fusion** (**AtCAF**) (Huang et al., 2025): AtCAF employs a counterfactual cross-modal attention module to identify causal relationships within the training data, thereby building a complete causality chain that tracks causal pathways from user inputs to model outputs. (7) **Disentangled-Language-Focused Model** (**DLF**) (Wang et al., 2025): It introduces a feature disentanglement module to distinguish between modality-shared and modality-specific information. Additionally, four geometric measures are incorporated to refine the disentanglement process, which helps to reduce redundancy and bolster language-targeted features. Moreover, a language-focused attractor is designed to enhance language representation by utilizing complementary modality-specific information. (8) **Disentangled-Language-Focused Model** (**DLF**) (Wang et al., 2025): It designs a feature disentanglement module to distinguish between modality-shared and modality-specific information. In addition, four geometric measures are introduced to refine the disentanglement process, helping to reduce redundancy and bolster language-targeted features. Furthermore, a language-focused attractor is designed to enhance language features by utilizing complementary modality-specific information. (9) **Contrastive FEature DEcomposition** (**ConFEDE**) (Yang et al., 2023): It concurrently executes contrastive representation learning and contrastive feature decomposition to enhance the multimodal representation. (10) **Missing Modality Imagination Network (MMIN)** (Zhao et al., 2021): MMIN creates robust joint multimodal representations through cross-modal imagination, allowing it to predict any absent modality from the available modalities under various missing modality scenarios. (11) **Causal Inference Distiller (CIDer)** (Zhong et al., 2025): CIDer primarily comprises two essential components: a Model-Specific Self-Distillation (MSSD) module and a Model-Agnostic Causal Inference (MACI) module. MSSD boosts robustness against feature missing by implementing a weight-sharing self-distillation method across low-level features, attention maps, and high-level representations. To address out-of-distribution issues, MACI utilizes a tailored causal graph to alleviate label and language biases via a multimodal causal module and fine-grained counterfactual texts. (12) **Graph Complete Network (GCNet)** (Lian et al., 2023): GCNet addresses the missing modality problem in conversational settings by incorporating Speaker GNN and Temporal GNN to capture speaker and temporal relationships. It simultaneously optimizes classification and reconstruction tasks to make use of both complete and incomplete modality data. (13) **Incomplete Multimodality-Diffused emotion recognition (IMDer)** (Wang et al., 2023): IMDer employs a score-based diffusion model to reconstruct missing modalities by converting Gaussian noise into modality-specific distributions. It utilizes the available modalities as conditional guidance to maintain consistency and semantic alignment throughout the recovery process. (14) **Mixture of Modality Knowledge Experts (MoMKE)** (Xu et al., 2024b): MoMKE adopts a two-stage framework: initially, unimodal experts are trained separately, and subsequently, they are jointly trained with both unimodal and joint representations. A Soft Router is incorporated to dynamically combine these representations, facilitating enriched and adaptive multimodal learning in scenarios with incomplete modalities. (15) **Noise Intimation based Adversarial Training (NIAT)** (Yuan et al., 2024): NIAT formulates modality imperfection with the modality feature missing at the training period and improves the robustness against various potential imperfections at the inference period.

The additional baselines for the MHD and MSD tasks include:

(1) **Multimodal Global Contrastive Learning** (**MGCL**) (Mai et al., 2023c): MGCL performs supervised contrastive learning on multimodal representations and develops various methods to create positive and negative samples for each representation. (2) **Multimodal Correlation Learning** (**MCL**) (Mai et al., 2023a): MCL constructs a supervised multimodal correlation learning task to preserve modality-specific information and obtain a more discriminative embedding space. (3) **Decoupled Multimodal Distillation** (**DMD**) (Li et al., 2023): To boost the discriminative features of each modality, DMD enables flexible and adaptive cross-modal knowledge distillation. This process splits each unimodal representation into modality-irrelevant and modality-exclusive spaces, and utilizes a graph distillation unit to handle each separated component in a more specialized and efficient way. (4) **Subject Causal Intervention** (**SuCI**) (Yang et al., 2024): It introduces a straightforward yet effective causal intervention module to separate the influence of subjects as unobserved confounders, thereby attaining unbiased predictions through genuine causal effects. (5) **Humor**

**Knowledge Enriched Transformer** (**HKT**) (Hasan et al., 2021): HKT is a promising approach for MHD and MSD, leveraging humor-centric features as external knowledge to tackle the ambiguity and sentiment information concealed within the language modality. (6) **Modality Order-driven module for Sarcasm detection** (**MO-Sarcation**) (Tomar et al., 2023): It incorporates a modality order-driven fusion module into a transformer network, enabling the ordered fusion of modalities.

The compared MLLMs include:

(1) **Humanomni** (Zhao et al., 2025): HumanOmni represents a sophisticated multimodal framework designed for the analysis of videos that focus on human elements. This model's design incorporates three specialized streams for assessing facial expressions, body movements, and social interactions. HumanOmni employs the SigLIP (Zhai et al., 2023) algorithm to process visual data, relies on Qwen2.5 (Hui et al., 2024) for managing linguistic inputs, and uses Whisper-large-v3 (Radford et al., 2023) for handling audio signals. Furthermore, it applies the MLP2xGeLU (Li et al., 2024a) architecture to convert audio features into text, thus enabling a seamless blend with visual and textual information.

(2) **Qwen2.5Omni** (Xu et al., 2025): Qwen2.5-Omni functions as a comprehensive multimodal platform capable of managing a variety of data formats including text, audio, and video, and is capable of generating both written responses and spoken words. This system is designed based on the Thinker-Talker model. The Thinker module analyzes textual, acoustic, and visual content to generate sophisticated representations along with corresponding text. Following this, the Talker module takes these complex representations and transforms them into a series of spoken language elements. Underpinning the system is a language model that begins with the foundational parameters of the Qwen2.5 model (Hui et al., 2024). For processing audio inputs, it incorporates a Whisper-large-v3 encoder (Radford et al., 2023), while video inputs are processed using the Vision Transformer (ViT) (Han et al., 2022) architecture.

(3) **MiniCPM-o** (Yao et al., 2024): MiniCPM-o, an open-source MLLM crafted by OpenBMB, is designed to handle inputs from images, text, audio, and video, and to produce superior text and speech outputs in a seamless end-to-end process. MiniCPM-o leverages the capabilities of SigLip-400M (Zhai et al., 2023), Whisper-medium-300M (Radford et al., 2023), and Qwen2.5-7B-Instruct (Hui et al., 2024), amalgamating a substantial parameter count of 8 billion.

(4) **VideoLLaMa2-AV** (Cheng et al., 2024): VideoLLaMA2-AV is designed to enhance the representation of spatial-temporal dynamics and the understanding of audio in video-audio tasks. VideoLLaMA2-AV integrates a spatial-temporal convolutional interface to effectively seize the intricate spatial and temporal details present in video content. It functions through a two-branch system encompassing a vision-language pathway and an audio-language pathway. The language decoders are initialized with parameters from the Qwen2 model (Team, 2024). The vision-language pathway utilizes the CLIP (ViT-L/14) model (Radford et al., 2021) as its primary vision component, processing video frames sequentially. In contrast, the audio-language pathway extracts audio characteristics through the application of BEATs (Chen et al., 2022).

