# OpenReview forum: "Addressing Missing and Noisy Modalities in One Solution: Unified Modality-Quality Framework for Low-quality Multimodal Data"
_ICLR.cc/2026/Conference — ICLR 2026 Conference Withdrawn Submission_

### Official Review · Reviewer_qgnE · 2025-10-31

**Soundness:** 3
**Presentation:** 3
**Contribution:** 2
**Rating:** 6
**Confidence:** 3

**Summary:**

The paper jointly tackles robustness to both missing and noisy modalities in multimodal affective computing (MAC). It proposes a Unified Modality-Quality (UMQ) framework with three key pieces:
- (i) a quality estimator trained using explicit anchors and a rank-guided pairwise loss so the model learns ordinal quality without relying on brittle absolute labels;
- (ii) a quality enhancer that “decouples” each unimodal feature into sample-specific and modality-specific parts;
- (iii) a Modality-Quality-aware Mixture-of-Experts (MQ-MoE) with routing and regularizers that encourages consistent expert assignment under similar quality configurations.

Empirically, UMQ matches or exceeds recent robust baselines under complete inputs, outperforms methods like MMIN, GCNet, IMDer, MoMKE under missing modalities (varying missing rates), improves over C-MIB and Multimodal Boosting under noisy modalities (Gaussian corruption of features).

**Strengths:**

- **Originality / Idea quality**. Casting both missing and noisy modalities as "low-quality" and learning an ordinal quality estimator with explicit anchors plus rank-guided training is thoughtful; it avoids brittle absolute labels and fits naturally with routing.
- **Architecture design**. The decoupling into sample-specific vs. modality-specific subspaces and the quality enhancer that borrows cross-modal sample-specific information are intuitive and empirically helpful via illustrated experiments.
- **Empirical breadth**. Competitive or SOTA results are shown (i) in complete-modality; (ii) for missing modalities across a range of missing rates; (iii) for noisy modalities, indicating task generality.

**Weaknesses:**

- **Concerns about high-quality anchors**.  The "highest-quality" anchor uses a low unimodal predictive loss threshold. This might not be a robust indicator, as it may risk quality to reflect ease of the task label for that modality, or potentially rewarding label-leakage or majority cues rather than modality fidelity.
- **Noise realism**. The paper primarily uses Gaussian feature corruption and treats missing as "extreme noise". In fact, degradation could come from misalignment (e.g. between audio and video signal – which is quite popular in MAC applications), video frame drop, …
- **Potential misuse of Mutual Information terminology**. I believe the constraint presented is a distance/similarity regularizer, not an MI estimator/variational bound. The authors should clarify this terminology usage.
- **Limited applications**. While some ideas are interesting and parts of the architecture are cleverly designed, the authors choose to apply the method on MAC applications. This is somewhat restrictive for the applicability of the method. I suggest the Authors add an adaptation or provide guidance to extend the pipeline to other tasks.

**Questions:**

Please refer to Weaknesses.

---

### Official Review · Reviewer_ogxu · 2025-11-03

**Soundness:** 2
**Presentation:** 1
**Contribution:** 2
**Rating:** 2
**Confidence:** 3

**Summary:**

The paper attempted to propose a unified framework for handling both noisy and missing modalities in multimodal learning. The proposed unified modality-quality (UMQ) archives this goal by the following three modules: quality estimator, quality enhancer, and modality-quality-aware mixture of experts. The proposed approach is evaluated on several multimodality learning tasks that combine information from 3 input modalities, including visual, acoustics, and language.

**Strengths:**

1. UMQ outperforms the compared methods on the benchmarks.

2. The concept of developing a unified approach to handle both noisy and missing modalities is technically sound and addresses an important problem

**Weaknesses:**

1. Overall, although the proposed framework reports state-of-the-art performance, it appears rather complex, with multiple tightly coupled components and hyperparameters. Beyond the reported results, it remains unclear whether this work meaningfully advances the field or stimulates further discussion within the research community.

2. The design of the UMQ framework is not theoretically grounded. Its design lacks an in-depth mathematical justification or toy-example simulation on how and why each component is important for the final framework.

3. I had a hard time going through the content of this paper and have several comments on the presentation of the paper: 1) The text within the figures is overly dense and too small to read clearly. 2) The paper is difficult to follow overall, especially in the algorithm section, where equations are presented in a cluttered manner and frequently referenced back to the main text. This disrupts the reading flow and makes comprehension challenging.

**Questions:**

- Line 218: Since this is a learnable parameter, what ensures that the bias embedding term effectively compensates for the error? Are there any experiments or simulations that validate this claim?
- How are the predefined hyperparameters $\eta$, $m$, and $\gamma$ (line 292) selected? More generally, how many hyperparameters does the proposed framework involve in total, and what strategy is used for their selection?
- What is the exact implementation of the quality enhancer EH operator in Eq. 8?
- What is R in Eq. 10?
- In Eq. 17, what is the difference between L_m^{est} and L_m?
- In Eq. 20, how is $\alpha_m$ constructed?

---

### Official Review · Reviewer_U1S7 · 2025-11-05

**Soundness:** 3
**Presentation:** 1
**Contribution:** 2
**Rating:** 2
**Confidence:** 3

**Summary:**

This paper introduces a Unified Modality-Quality (UMQ) framework comprising three key components that jointly address the challenges of missing and noisy modalities in multimodal affective computing. Extensive experiments on five benchmark datasets—CMU-MOSI, CMU-MOSEI, CH-SIMS, UR-FUNNY, and MUStARD—demonstrate that UMQ consistently outperforms prior methods under both complete and degraded modality conditions.

**Strengths:**

1. Comprehensive experiments: The authors evaluate the approach under multiple conditions (complete, missing, and noisy modalities) and across several datasets.

2. Ablation studies and visualizations: They carefully analyze the effect of each component (estimator, enhancer, MQ-MoE) and visualize the improvement qualitatively.

**Weaknesses:**

Major:

1. The manuscript is poorly written, with awkward phrasing and overly long sentences (e.g., in the abstract) that obscure the main ideas. Clarity is a major problem throughout.

2. The paper lacks a formal definition or analysis showing why missing modalities can be treated as a subclass of noisy modalities from an information-theoretic or probabilistic perspective.

3. Besides manually added noise, can the proposed method also handle naturally occurring ones, such as those caused by poor devices?


Minor:

Multiple formulas are stacked together (e.g., 2–5) and explained with excessive text, while key model construction details are placed in the appendix. This structure obscures the main idea.

**Questions:**

Please think carefully about the issues mentioned in the **weaknesses** section.

---

### Note · Authors · 2025-11-13

I have read and agree with the venue's withdrawal policy on behalf of myself and my co-authors.